# Health-Promoting Capacities of *In Vitro* and Cultivated Goji (*Lycium chinense* Mill.) Fruit and Leaves; Polyphenols, Antimicrobial Activity, Macro- and Microelements and Heavy Metals

**DOI:** 10.3390/molecules25225314

**Published:** 2020-11-14

**Authors:** Arleta Kruczek, Marcelina Krupa-Małkiewicz, Sabina Lachowicz, Jan Oszmiański, Ireneusz Ochmian

**Affiliations:** 1Department of Horticulture, West Pomeranian University of Technology Szczecin, Słowackiego 17 Street, 71-434 Szczecin, Poland; ka21499@zut.edu.pl; 2Department of Plant Genetics, Breeding and Biotechnology, West Pomeranian University of Technology Szczecin, Słowackiego 17 Street, 71-434 Szczecin, Poland; Marcelina.Krupa-Malkiewicz@zut.edu.pl; 3Department of Fermentation and Cereals Technology, Wrocław University of Environmental and Life Science, 37, Chełmońskiego Street, 51-630 Wroclaw, Poland; sabina.lachowicz@upwr.edu.pl; 4Department of Fruit, Vegetable and Plant Nutraceutical Technology, Wrocław University of Environmental and Life Science, 37, Chełmońskiego Street, 51-630 Wroclaw, Poland; jan.oszmianski@upwr.edu.pl

**Keywords:** antioxidant, antibacterial activity, mineral elements, *in vitro*, fruit, leaves

## Abstract

There is a growing interest among the public in fruit with a positive impact on human health. Two goji berry cultivars (‘No. 1’ and ‘New Big’) were propagated *in vitro*, grown in an orchard and then evaluated for macro- and microelements and harmful heavy metals (i.e., Pb, Ni, and Cd). The leaves and fruit were also assessed for nutritional value, polyphenols and the antimicrobial activity of the leaves. ‘New Big’ was characterized by a higher content of macro elements in the leaves (*in vitro* and orchard) and a higher content of microelements in the fruit. The harmful substances content was below the minimum value. Furthermore, neither the fruit nor the leaves contained cadmium. This study also indicated that leaves had a higher content of polyphenols compared to the fruit. The fruits were characterized by their health-promoting capacities, while the leaves were characterized by their antibacterial activity. Among the Gram-positive bacteria, the most sensitive strain was *Bacillus subtilis*, and among the Gram-negative bacteria, it was *Proteus vulgaris*. Taking into consideration the Recommended Daily Allowance (RDA) for minerals, goji berries can be declared to be a source of Cu, Fe, Mn, Zn and P.

## 1. Introduction

The consumption of red berries has increased considerably in recent years. Werewolf berries (*Lycium chinense* Mill, *Solanaceae*), traditional food and medicines in East Asia, have become progressively more popular in Europe and North America [1,2,3] since the beginning of the 21st century. Two closely related species, *L. barbarum* and *L. chinenese*, are well known around the world, and both function as food and medicinal plants in China and other Asian countries. *L. chinense* Mill. berries contain multiple mineral and organic compounds (i.e., vitamins B_1_, B_6_, A, C, E), with potential to repair epidermal damage, and showing excellent effects on cardiovascular and cholesterol levels [3,4]. Goji leaves are also a rich source of bioactive compounds that can be used as additive in health-promoting preparation [5,6]. In support of such traditional properties, modern studies indicate that extracts from goji berries possess a range of biologic activities, including antioxidant properties [7]. According to Mocan et al. [5], goji fruits have an effect on ageing, neuroprotection, glucose control in diabetics, antioxidant properties, immunomodulation, and antitumor activity, as well as general well-being, fatigue, metabolism, energy expenditure, and cytoprotection. In addition, no harmful substances, such as NO_2_ and NO_3_, or the presence of steroids (α-solanine) and tropane alkaloids (skopolamin) have been reported in goji fruit [6]. According to Kulczyński and Gamza-Michałkowska [8], a well-balanced diet and a change in lifestyle have significant impacts in terms of reducing the frequency of diseases in society. Thus, natural antioxidants, particularly in fruits and vegetables, have gained interest among consumers and the scientific community.

Goji shrubs are propagated vegetatively, especially by semi-wooden cuttings and with great success *in vitro* [9]. According to Dzhugalov et al. [10], the optimum quality fruit can be obtained under hot summer conditions. Rain causes fruit cracking during ripening. Thus, there is a relationship between environmental conditions and the harvest. Optimal growing conditions allow for a production of almost 95,000 tons of fruit annually from August to October. Goji fruit is usually dried, but both fruit and young leaves, which are a valuable source of mineral compounds, can also be eaten raw [1,11]. In addition, fruit extracts are used as natural, non-toxic colorants for drinks or cosmetics [12]. In the literature, the main focus has been on the properties of goji fruit. However, the leaves of *L. chinense*, called “tianjingcao” (vitality and vigor of nature) have rarely been studied, despite their use in traditional Chinese medicine in alleviating mineral deficiency, combating heat distress, quenching thirst, dispelling wind, and enhancing eyesight. Moreover, *L. chinense* leaves have also been widely used as tea, medicinal vegetables, and herbal drugs in China [5]. Goji leaves contain high amounts of specific flavonoids and phenolic acids, such as chlorogenic acid, quercetin, and rutin. Additionally, goji leaves contain polysaccharides that exert high superoxide and DPPH scavenging abilities, thereby having high antioxidative activity [13]. Information regarding goji leaves, including their chemical composition and bioactive features, is not comprehensive and quite limited. However, recently, *L. chinense* leaves have been described as a sustainable source of antioxidants and antibacterial compounds.

Considering the excellent health properties and wide use of goji fruit and leaves, we compared the physical-chemical properties of goji leaves and fruit grown in orchard and *in vitro* culture. In addition, the particular aim of this study was to assess the antimicrobial properties of goji leaves as a scientific basis for the further use of this plant as a functional food/ingredient.

## 2. Results and Discussion

### 2.1. Mineral Compound of Leaves and Goji Fruits

N, P, K, Ca, Mg and Na (i.e., macroelements) are important for several physiological functions. Each of the ingredients plays a specific role in a plant’s metabolism. The most important macroelement is nitrogen, a component of amino acids and proteins, DNA and RNA nucleic acids, enzymes, hormones, and energy carriers. Sodium, for example, is a macroelement responsible for membrane depolarization and, water control, and is a physiological cotransporter. Magnesium is required for nucleic acids in protein synthesis. The sodium and magnesium values in dried fruits correspond to 1/5 of the daily human need [14].

The contents (g/kg) of six macroelements in the leaves of two goji (*L. chinense* Mill.) cultivars in the samples analyzed are shown in Table 1. The distribution of the compositions of minerals in the different parts of the plant (i.e., fruit or leaves) was determined by the cultivation conditions. In general, the elemental concentrations in all samples decreased in the following order: N > K > Ca > P > Mg > Na and N > K > P > Ca > Mg > Na in the leaves of both cultivars, respectively, from the orchard and *in vitro*. Among the analyzed macroelements, the highest amount of nitrogen was in the leaves. This high level was found in both varieties grown in the orchard, as well as under the *in vitro* conditions (Table 1). However, the contents of macroelements such as N, P, and K were significantly higher in plants propagated *in vitro* compared to plants from the orchard cultivation, despite the high content of these components in the soil. Conversely, the contents of Ca, Mg, and Na were significantly lower in both cultivars *in vitro* compared to orchard conditions. The leaves from the New Big cultivar accumulated significantly higher concentrations of N, P, Ca, Mg, and Na. The exception was potassium, which in cultivar No. 1 was higher by 50% in the orchard and 9% *in vitro*. The mineral content of the leaves is also an indicator of a plant’s nutrition and condition. There is no information available in the literature about the optimal nutrient content in goji leaves. In this experiment, the recommended values for goji were compared to berry shrubs [15,16]. It was found that in the leaves, regardless of growth conditions, there were very low level of P and Ca. The remaining macroelements were in the optimal range or even exceeded it.

Fe, Zn, Mn, Cu and Se (i.e., microelements) perform an important function in many biochemical reactions. They are cofactors in enzymatic reactions, such as glucose tolerance factor, and metallo-enzymes in several endogenous reactions, including insulin storage, the immune system, and hormone activators [14,17]. The values obtained for five microelements (i.e., Fe, Zn, Mn, Cu, Se) in the leaves of the two goji cultivars are shown in Table 2. In general, the concentrations of the microelements in all samples decreased in the following order: Fe > Mn > Zn > Cu > Se > Pb > Ni in the orchard, and Mn > Fe > Zn > Cu *in vitro*. It was observed that in leaves of the No. 1 cultivar, these values were higher in comparison to in those of the New Big cultivar. This was opposite to the plants propagated *in vitro.* Moreover, no heavy metals such as Pb, Ni, and Cd were found in the leaves of the plants propagated *in vitro*. This was due to the absence of these elements in the MS medium. Additionally, in the leaves of goji from the orchard, no Cd was found, although its content in the soil was 0.296 mg/kg. Moreover, ‘New Big’ in the orchard variety accumulated 38% more lead in the leaves, while the ‘No. 1’ accumulated 72% more nickel in the leaves. This condition can be explained as being due to the mineral variations in the soil and medium, as well as a lower bioaccumulation of these minerals in the leaves. The growing conditions were not the same. Similar to the macroelements, the optimal content of the microelements was compared with the standards for berry shrubs [15,16]. Therefore, it was found that in the leaves collected in the orchard, only Mn was at a low level.

The leaves of both tested cultivars of goji contained greater amounts of macro- and microelements and metals than the fruit. Although the leaves were contaminated with more than twice the amount of lead and nickel than the fruit, this was attributed to the orchard being situated close to an industrial site, and therefore potentially being subject to heavy air pollution. Lead is more specific to air pollution than cadmium, and we did not detect it in neither the leaves or the fruit. However, the fruit of the No. 1 cultivar contained 41% less lead and 33% more nickel. These values were higher than those obtained by Sá et al. [14] in goji fruit grown in South America and those of Kulaitienė et al. [18] in goji fruit grown in Lithuania. Jeszka-Skowron et al. [19] showed that the presence of Cd was 0.046 mg/kg, Pb was 0.109 mg/kg, and Ni was 2.61 mg/kg in goji fruit. Llorent-Martínez et al. [20] established that Cd, Pb, and Ni in goji fruit was 0.035–0.090 mg/kg, 0.035–0.095 g/kg, and 0.33–0.90 mg/kg, respectively. According to the Regulation of the European Commission [21], the values defined for fruit by law are 0.05 and 0.20 mg/kg for Cd and Pb, respectively. In the present study, the concentrations of toxigenic elements (i.e., Pb and Ni) were low and below the permissible limit levels.

The contents of macroelements in the goji fruit obtained in this study differed from those reported by other authors [2,14,18,22]. We did not expect the contents of the studied macroelements to be identical. It was quite difficult to find data for comparison of the same type of leaves and fruit that were investigated in this study. The major macroelements were as follows: N > P > K > Na > Mg > Ca for both cultivars (Table 1). Nitrogen is a predominant element (24.32–29.85 g/kg) for the goji fruit of cvs. New Big and No. 1. It was observed that cv. No. 1 had a significantly higher content of macroelements compared to cv. New Big. Significant differences were found between the cultivars in their concentrations of N, P, Ca, and Na. An opposite relationship was observed in the case of the concentrations of microelements in the studied goji fruit (Table 2), in which the major microelements were as follows: Fe = Cu > Zn > Mn > Se. Ferrum (i.e., iron) was found in the highest concentration in the goji fruit (66.03–79.44 g/kg, with the highest levels in ‘No. 1’ followed by ‘New Big’). The New Big cultivar was characterized by a significantly higher content of microelements in its fruit, except for selenium, which was 25% lower than in the fruit of the No. 1 cultivar.

Our results differ from those obtained by Sá et al. [14], where the concentration of macroelements in goji fruit was within the limits of 3.124–590.7 mg/kg. Additionally, Nascimento et al. [2], Llorent-Martínez et al. [20], and Niro et al. [22] reported slightly different results. As in other plant food, the mineral content of berries reflects the soil in which they are grown. It should be highlighted that the content of required and undesirable elements in the fruit depends largely on the properties of the soil, plant physiology, source and chemical composition of water, fertilizers, pesticides, insecticides, and fungicides used on the plantation. Each species has individual requirements and different tolerance levels to the absorbed and accumulated chemical compounds from the medium [22].

The study of the chemical composition of fruits that are considered to be superfoods is important from nutritional and toxicological perspectives. The percentages of macro- and microelements in the goji fruit were compared to the daily mineral requirements of the human diet [21]. Recommended Daily Allowances (RDAs) have been established, namely, 2 mg of Cu, 18 mg of Fe, 400 mg of Mg, 2 mg of Mn, 1000 mg of P, and 15 mg of Zn [2]. Niro et al. [22] reported that 100 g of fresh goji berries can be declared on the label as a source of Cu. The consumption of 100 g of fresh goji per day contributes to approximately 25% of the RDA of Cu [22]. In our study, we obtained similar results, where 100 g of fresh goji fruit constitutes 30% and 35% of the demand for this element, respectively, for the No. 1 and New Big cultivars (Table 2). Moreover, fresh goji fruit is also a rich source of ferrum. One hundred grams constitutes 47% of the RDA for this element; this was also confirmed by our own research. Copper is involved in maintaining cardiovascular health, and glucose and cholesterol metabolism [23]. Iron is an essential micronutrient due to the fact of its high functionality, while also being an important functional food for patients with iron deficiency. In contrast to the results obtained for goji fruit by Niro et al. [22], the fruit we studied were also a small source of selenium. In fact, 100 g of fresh berries contributed to approximately 13% and 17% of the RDA, respectively, for the New Big and No. 1 cultivars. In connection with the use of goji fruit in disease-preventing diets, the results of this study are interesting and very helpful and important. Knowledge of minerals and trace elements in natural products is extremely important. Goji fruit, called functional foods may be a main external source of these nutrients for humans and animals alike.

The analysis carried out using the Ward’s method (Figure 1a) identified three groups with similar micro and macroelements in goji. This may indicate that the content of mineral components in leaves and fruits was more dependent on the place of cultivation (*in vitro*-group a, or orchard-group b). A completely separate group, connected with leaves in a very weak way, was the fruit-group c.

### 2.2. Leaf and Fruit Color

According to Kiełbasa and Juliszewski [24], by measuring the area of one leaf, we can determine the intensity of growth and the related intensity of the photosynthesis process. Based on these findings, we can assess the condition of the plants [25]. In this study, the leaf surfaces of the two goji cultivars grown in the orchard and *in vitro* were compared (Table 3). Upon analyzing the leaves, it was found that the New Big cultivar had the largest leaf area, with a size varying from 4.18 cm^2^ (orchard) to 2.78 cm^2^ (*in vitro*) in comparison to the No. 1 cultivar. The results obtained were similar to those described by Kruczek et al. [6]. The leaves of the plants grown *in vitro*, despite the fact that they had more mineral components in their composition, that is, they were better nourished, which was also indicated by their more intense color; they were, however, much smaller.

The results of the color determination of the leaves and fruit were also analyzed in the transmitted mode using the photocolorimetric method in the CIE L**a*b** system. The leaves of *L. chinense* Mill. were also characterized by a similar value of the green index [3,6]. The value of the *a** parameter (color ranging from green to red) determined on the surface of leaf ranged from −33.84 to −43.56 (Table 4). The ‘New Big’ goji leaves under *in vitro* conditions were 22% greener compared to the goji leaves grown in the orchard. This could have been influenced by the much higher content of most macro- and microelements in these leaves (Table 1 and Table 2). However, the value of the *a** parameter in the case of the No. 1 cultivar was at a similar level regardless of the cultivation conditions. The leaves of the New Big cultivar were also darker, which is shown by the lower value of the parameter L* (34.57 and 42.50, *in vitro* and orchard, respectively). The value of parameter L* (reaching from 0 to 100, black to white, respectively) is usually used for tracking color changes [6,26]. However, the goji leaves of cultivar No. 1 were 23% (*in vitro*) and 13% (orchard) brighter compared to the ‘New Big’ leaves. The color of the leaf surface was determined by parameter *b** (from a yellow to a blue color) which ranged from 23.71 (‘New Big’ *in vitro*) to 30.63 (‘No. 1’ orchard). These results are consistent with the Normalized Anthocyanin Index (NAI) (−1 and +1, lack of redness and red, respectively). Leaves that were intensely green had the lowest NAI index and ranged from -0.69 (‘No. 1’ *in vitro*) to −0.82 (‘New Big’ orchard). On the other hand, the Normalized Difference Vegetation Index (NDVI), which indicates the vegetative potential of plants, was nearly 10% higher in the goji leaves propagated *in vitro* (Table 3). These results obtained are comparable to those described by Kruczek and Ochmian [3], and Kruczek et al. [6].

The color of fruit is of crucial importance in consumers’ choices. An intensive color may encourage purchases, and/or discourage consumption by warning of their rottenness [6,27,28]. According to Ochmian et al. [26], the color of fruit depends on the place of cultivation and the climatic conditions prevailing there. The color of food products may also change during processing [29]. Regarding the testing of the fruit color, the values of the *a** parameter measured were significantly different. The No. 1 cultivar fruit had more than a 23% red-coloring compared to the ‘New Big’ fruit (Table 3). Parameter *a** was corelated with the NAI [6,26]. The content of anthocyanin pigments determined by the NAI value was responsible for the color of the goji fruit. The NAI showed that the highest anthocyanin contents were recorded in the fruit of the New Big cultivar. When estimating the ripeness and quality of the fruit, the NDVI value was also determined, which in the tested goji fruit, was −0.38 in ‘New Big’, indicating immature fruit (pre-climacteric). In cultivar No. 1 the value of the NDVI index was −0.46 and characterized fruit with the onset of ethylene production. The obtained results were higher than those described by Kruczek et al. [6].

The color of the surface of the goji fruit skin was described by parameter *b**. The New Big cultivar contained the highest quantity of red-colored substances (41.40) and a smaller quantity of yellow-colored ones, and these values were in accordance with the values obtained by Kruczek et al. [6]. The value of the L * parameter was 20% higher in the New Big cultivar in comparison to the No. 1 cultivar. The results obtained in this study were compared to the results of Lachowicz et al. [28] in saskatoon berries (L* from −32.71 to 49.31) and Kruczek et al. [6] in goji berries (L* from −32.6 to 72.3). However, the goji fruit examined by Kruczek and Ochmian [3] and the highbush blueberries examined by Ochmian et al. [30] were darker than in our own research.

### 2.3. Polyphenolic Compounds and Health Promoting Capacities

Polyphenols are secondary metabolites of plants and are generally involved in defense against biotic and abiotic stresses. The content of polyphenols in plants is influenced by many factors, including the degree of maturity at harvest, environmental factors, processing, and storage. In the last decade, there has been great interest in the potential health benefits of dietary plant polyphenols as antioxidants, identifying 10 flavonols, two flavan-3-ols, two tannins, and 10 phenolic acids (Table 4). The two cultivars of goji grown *in vitro* provided similar values in terms of total phenolic content (102.64 mg/100 g DW and 103.92 mg/100 g DW, ‘No.’1 and ‘New Big’, respectively), while orchard-grown goji had a significantly higher amount (124.46 mg/100 g DW and 167.79 mg/100 g DW, ‘No. 1’ and ‘New Big’, respectively). This finding indicates that goji species are a significant source of phenolics. According to Wojdyło et al. [31], the average content of phenol in all goji cultivars grown in Poland was 97.23 mg/100 g, with considerable inter-cultivar differences. The content of total polyphenolic compounds was statistically different among the different species. Kiwifruit and apple contained small quantities of polyphenolic compounds (70.23–83.40 mg GAE/100 g FW), while a significantly higher polyphenolic content was observed in strawberry (323.39 mg GAE/100 g FW), and blackcurrant (434.43 mg GAE/100 g FW). The polyphenol content in the tested cultivars of goji was between that of orange (158.70 mg GAE/100 g FW) and guava (310.10 mg GAE/100 g FW). In this study, the dominant phenolic acid was 5-*O*-caffeoylquinic acid.

According to Sato et al. [32] and Kruczek et al. [6], chlorogenic acid has a significant influence on the flavor of fruit and vegetables. Moreover, it shows anticarcinogenic, antimutagenic, and antioxidant properties *in vitro*. In the human body, chlorogenic acid is poorly absorbed and metabolized by colonic microflora [32]. In this study, a significantly higher concentration of chlorogenic acid was detected in the leaves grown in the orchard than *in vitro* (Table 4). This is in accordance with the results obtained by Chen et al. [13], who also confirmed that in goji fruit the concentration of chlorogenic acid is very low in comparison to the leaves. This demonstrates that goji leaves are a valuable source of chlorogenic acid

Flavonoids, belonging to polyphenolic compounds, are also commonly found in plants, especially in fruit and vegetables [33]. The total amount of flavonoids in the flavonoid fraction was higher in the goji leaves than in the fruit. Moreover, quercetin-3-*O*-Gal and quercetin-3-*O*-Rut-7-*O*-Glu, which were present in the fruit, were not identified in the leaves of both cultivars. On the contrary, kaempferol-3-*O*-Glc-7-*O*-Soph, quercetin-3-*O*-Rut-7-*O*-Rha, and kaempferol-3-*O*-Glc-7-*O*-Rha were only identified in the leaves of both cultivars (Table 5).

For human health, gallotannins are essential [6]. The highest content of tetragalloyl-glucose, regardless of the cultivation method, was found in ‘New Big’ (4.22 and 3.81 mg/g DW, *in vitro* and orchard, respectively). In ‘No. 1’, these values were 95% and 92% lower *in vitro* and in orchard, respectively.

According to many authors [6,33,34], the higher the content of total polyphenols, the higher the antioxidant activity. Pandey and Rizvi [34] suggested that the long-term use of diets rich in plant polyphenols provides some protection against the development of cancer, cardiovascular diseases, diabetes, osteoporosis, and neurodegenerative diseases.

### 2.4. Antioxidant Activity

The antioxidant activity was evaluated using DPPH scavenging activity and ferric-reducing antioxidant power (FRAP). Free radicals are known to be a major factor in biological damage. The DPPH radical-scavenging assay is a widely used method to evaluate the ability of plant extracts to scavenge free radicals generated from the DPPH reagent. The DPPH free radical scavenging activity of the two tested goji cultivars is presented in Table 5. Relatively higher DPPH scavenging abilities were recorded in goji fruit (7.61 mmol Trolox/100 g and 5.33 mmol Trolox/100 g, ‘No. 1’ and ‘New Big’, respectively), while the lowest DPPH scavenging abilities were found in the leaves grown in the orchard (3.26 mmol Trolox/100 g, 2.48 mmol Trolox/100 g, ‘No. 1’ and ‘New Big’, respectively).

The analysis carried out using the Ward method (Figure 1b) showed that the polyphenol content was divided into three groups with similar micro- and macroelements in goji. The fruits formed a separate group (b), and similar for the leaves of New Big cultivar (c).

Regarding the values of the total antioxidant capacity, expressed as the FRAP assay, the results showed large statistical variations among the tested cultivars. The highest radical scavenging activity (FRAP) was obtained in the goji leaves grown *in vitro* (4.02 mmol Trolox/100 g and 5.89 mmol Trolox/100 g, ‘No. 1’ and ‘New Big’, respectively); meanwhile, the FRAP values obtained for the leaves grown in the orchard and for fruit were at a similar level (Table 5). Our data agree with that reported by Kruczek et al. [6] and Wojdyło et al. [31], who demonstrated similar values of antioxidant activity in their studies. According to many authors [6,26,31], the antioxidant activity of various plant species (fruit or leaves) may be determined by growing conditions, geographical location, climatic conditions, genotype, fruit maturity, or even the collection methods. Many authors have highlighted that fruit rich in significant amounts of phytochemicals are of great interest to potential consumers [6,13].

### 2.5. Antidiabetic Activity of Goji Fruits

The inhibition of *α*-amylase in the analyzed leaves ranged from 75.1 (‘No. 1’ in the orchard) to 172.9 mg/mL IC_50_ (‘New Big’ *in vitro*), and in the fruits ranged from 33.45 mg/mL IC_50_ in ‘No. 1’ to 37.01 mg/mL IC_50_ in ‘New Big’. While the inhibition of *α*-glucosidase in leaves samples was between 22.05 (‘No. 1’) and 42.28 mg/mL IC_50_ (‘New Big’). Goji leaf extracts collected *in vitro* were more effective in the activity of *α*-amylase and *α*-glucosidase inhibitors (Table 5). However, there were no statistical differences in the activity of *α*-amylase and *α*-glucosidase in fruits. The inhibition of these enzymes may be effective for regulating type 2 diabetes by controlling the absorption of glucose [35]. Both α-amylase and α-glucosidase inhibitory activity in fruits such as pomegranate, strawberry, raspberry, pear, kiwi, plum, lingonberry, black currant, and blueberry extracts has been confirmed in other studies [16,28,31,35]. Inhibition of these enzymes is specifically useful in the treatment of non-insulin-dependent diabetes, as it slows down glucose release into the bloodstream [31].

### 2.6. Soluble Sugars and Organic Acid

The organoleptic properties of the fruit are influenced by many ingredients, including sugars and organic acids [31]. The sugar content in the goji samples examined in this study is presented in Table 5. The main sugars identified in the analyzed leaves and fruit of the two cultivars of goji were fructose, glucose, and sucrose. The largest amounts of total sugars were fructose and glucose, both in the leaves (0.89–2.51 g/100 g DW and 0.51–1.88 g/100 g DW, fructose and glucose, respectively) and the fruit (9.67–11.83 g/100 g DW and 10.11–13.06 g/100 g DW, fructose and glucose, respectively). The cultivar with the highest sugar content in both its leaves and fruit was ‘New Big’. Moreover, the leaves from goji grown *in vitro* had from 80% to a 120% higher total sugar amount than in the leaves grown in the orchard. This may be due to the addition of 3% sucrose to the MS medium.

Our results show significant differences (*p* < 0.05) in the content of organic acids among the tested cultivars (Table 5). The total organic acid content ranged from 0.182 to 0.450 g/100 g DW in the leaves and from 1.576 g/100 g DW in ‘No. 1’ fruit to 2.235 g/100 g DW in ‘New Big’ fruit. The New Big cultivar had the highest amount of organic acid, and it was also observed that leaves from goji grown *in vitro* were the lowest. Oxalic, citric, succinic, and fumaric acids were the main organic acids in the leaves and fruit. However, in the fruit from ‘No. 1’, oxalic acid was not detected. The highest amount of total organic acid was found to be citric acid, both in the leaves and the fruit. Other organic acids, (i.e., oxalic, succinic, and fumaric) were present in low concentrations. The types and amounts of sugars and organic acids were the same as reported by Montensano et al. [36] and Wojdyło et al. [31]. According to Potterat [1], the content of polysaccharides and total sugars in *L. chinensis* fruit is a major medicinal aspect of goji berries. The sugar/acidity ratio in fruit is an important choice for consumers.

PCA for minerals and phytochemical content of leaves and fruits explained (70.94) the total variance, where PC1 represented 45.30% and PC2 25.64% (Figure 2). The statistical method pointed out three major parts. Our results show that there were many correlations (positive, negative, weak) between antioxidant capacity and mineral extractability. The first group included polyphenols, which showed high dependence with antioxidant activity, and inhibitors activities. It is known that the antioxidant activity of fruits is influenced by their phenolic composition [37]. There was a positive correlation between the DPPH values and polyphenolic compounds content, meaning that the concentration of phenolic compounds may be a good indicator of the reducing capacity in the fruits. Polyphenolic compounds such as phenolic acids, flavonoids, anthocyanidins, and tannins, produced as secondary metabolites by plants, possess remarkable antioxidant and immunomodulatory activities [38]. A high correlation between polyphenols and NDVI index was also found. It was also found that macroelements (especially K, P, and N) were negatively correlated with microelements. This may be due to synergism and antagonism between minerals. It is well known, for example, that high phosphorus content may limit the uptake of microelements. A high content of these elements was found in the soil (Table 7) and leaves (Table 1 and Table 2) of the studied goji cultivars. There is also antagonism between Cu and Zn, and potassium strongly limits the uptake of Na. In contrast, potassium shows strong synergy with nitrogen. There may also be an overaccumulation of some components in the soil (e.g., phosphorus and potassium) and depletion of others (magnesium and microelements).

In the second group, a correlation between FRAP and some microelements. Generally, positive correlations were observed between FRAP and Fe, Cu, and Sn content in our research. These results demonstrate that FRAP in fruits has a significant impact on enhancing the extractability of Cu bound with the ability to form chelated metal ions by FRAP and to retain them (37). Antioxidants that react in the FRAP assay are those that can reduce, under the reaction conditions used, the Fe^3+^-TPTZ salt to Fe2+-TPTZ form. These include polyphenolic compounds such as catechins and other flavonoids in plant-based foods [39].

In the third group, the dependency between NAI index and color parameters *a** and *b** was visible.

No anthocyanins were found in the polyphenolic profile of goji, as evidenced by the negative value of the NAI index (also a color indicator—Table 3). This is confirmed by high dependence on other parameters of color a * and b * and Mg, the basic component of chlorophyll. This correlation may indicate that the presence or absence of one metal has no effect on the other.

### 2.7. Antimicrobial Activity

Plants are an important source of potentially useful structures for the development of new chemotherapeutic agents. *In vitro* antimicrobial activity should be determined first [40]. The leaf extracts in the tested concentrations showed inhibitory activity (MIC) on the growth of the analyzed Gram-positive bacteria, especially on *S. aureus*. However, among the Gram-negative bacteria, only *P. vulgaris* was sensitive to its effects (Table 6). A definitely higher inhibitory effect characterized extracts from the leaves collected from the shrubs that grew in the orchard, especially from the New Big cultivar. The antimicrobial activity of the leaf extracts has been confirmed in studies conducted with the use of the disc diffusion method (Table 6). Extracts from the leaves harvested in the orchard of both studied cultivars had a higher inhibition zone (9.0–16.7 mm) compared to the *in vitro* leaves (8.6–14.2 mm). The smallest inhibition diameter was determined for Gram-negative *E. coli* (8.6 mm) and the largest (16.7 mm) for Gram-positive bacteria such as *B. subtilis*. However, these inhibition capacities are lower than those observed by Mocan et al. [5] and Dahech et al. [41].

## 3. Materials and Methods

### 3.1. Characteristics of the Research Area

#### 3.1.1. Orchard Experiment

The study was carried out in the Department of Horticulture and the Department of Plant Genetics, Breeding and Biotechnology, of the West Pomeranian University of Technology in Szczecin. The research station is located in subzone 7A in the North-Western part of Poland in the Szczecin Lowland at a distance of approximately 65 km from the Baltic Sea (53°40′ N, 14°88′ E). The research was conducted at a production plantation specializing in the cultivation of highbush blueberry, located in the Szczecin’s Lowland. In this area, there are numerous hills of 40–60 m a.s.l., the remnants of the frontal moraine. The climate of this area is also significantly affected by the presence of big water basins (Szczecin Lagoon, Dąbie Lake, the Odra River), which provide additional moisture in the period of plant vegetation. The average growing season (April–October) temperature from 1951 was 13.7 °C and rainfall 391 mm [42].

#### 3.1.2. *In Vitro* Experiment

The second part of the experiment was carried out in the Laboratory of *In vitro* Cultures, where all cultures were incubated in a growth room at a temperature of 24 ± 2 °C under a 16 h photoperiod with a photosynthetic photon flux density (PPFD) of 40 μmol/m^2^/s provided by Narva (Germany) emitting daylight cool white.

### 3.2. Characteristics of the Plant Material

#### 3.2.1. Orchard Experiment

The soil (pH 6.75) in the orchard was an agricultural soil with a natural profile, developed from silt loam (sand 42.7%, silt 52.9%, clay 4.4%) with a considerably lower density of 1.23 Mg/m^3^. The groundwater level was 140–160 cm, and a higher water capacity of 46.2% ww. It also contained much more organic matter—32.4 g/kg of soil. Regardless of the site, the soil was characterized by similarly low salinity EC 0.33–0.42 mS/cm. The mineral content of the soil is shown in Table 7. The soil, in which the shrubs grew, regardless of the stand, in comparison to the optimal mineral content of the soil by Sadowski et al. [43], was characterized by a high content of P, K, and Mg. Every spring nitrogen fertilization was used at a dose of 45 kg N. Irrigation of the plantation was carried out annually using a permanently installed T-Tape drip irrigation line with the emitter’s performance of 1.5 L/h (3 L of water on a section of 1 linear meter of the installation). The moisture content of the soil was maintained in the PF 1.8–2.1 range and was determined using contact tensiometers.

#### 3.2.2. *In Vitro* Experiment

The research material consisted of 15–20 mm stem nodes with an axillary bud of goji obtained from a sterile stabilized *in vitro* culture. The explants were transferred to MS medium according to Murashige and Skoog’s [44] composition of vitamins, and macro- and microelements. All media contained 30 g/dm^3^ of sucrose (Chempur, Poland) and 100 mg/dm^3^ of myoinositol (Duchefa, The Netherlands) and were solidified with 8 g/dm^3^ of agar (Biocorp, Poland). The pH of the media was adjusted to 5.7. The media were heated and then 30 mL was poured into 450 mL flasks, which were autoclaved at 121 °C (0.1 MPa) during the time required according to the volume of medium in the vessels. After the end of the experimental period (five weeks), the explants were removed and washed with deionized distilled water.

### 3.3. Analysis of the Chemical Compounds

For the chemical analyses, leaves were taken from the orchard at the beginning of August (during harvesting of the fruit); 100 pieces were taken from each combination. Typical, healthy leaves from the middle part of the annual shoots were collected. Leaves were taken from the multiplied 5-week shoot cultures of goji plants, to be used for chemical analyses. From each harvest, we also took samples of fruits that were then frozen. After finishing the harvest from each period, we prepared a collective sample that was dried (and 65 °C) and ground.

The content of elements in leaves and fruits were determined after mineralization: N, P, K, Ca, and Na were measured after wet mineralization in H_2_SO_4_ (96%) and HClO_4_ (70%). The content of Cu, Zn, Mn, Fe, Se were determined after mineralization in HNO_3_ (65%) and HClO_4_ (70%) in a ratio soil 1:1 [45], leaves and fruits 3:1 [46]. The total N concentration in plants was determined by the Kjeldahl distillation method [47]. The content of K was measured with the atomic emission spectrometry, Mg Ca, Na, Cu, Zn, Mn, Se, Fe Pb, Ni, and Cd content with the flame atomic absorption spectroscopy. P was assessed by the colorimetric method [16]. All tests were performed each year in three replications.

### 3.4. Nutritional Value and Polyphenols of Leaves and Fruits

Samples (leaves and fruit) were freeze-dried before the analysis, and then powdered and subjected to the extraction process according to the methodology of Oszmiański et al. [48]. Nutritional value was determined in the dehydrated and leaves and fruit. Soluble sugars and organic acids were determined according to the procedure described by Dias et al. [49]. Soluble sugars and organic acids were determined by UFLC-PDA.

The FRAP (Ferric-Reducing Antioxidant Power) assay was done according to Benzie and Strain [50] and the 1,1-diphenyl-2-picrylhydrazyl (DPPH) was done according to Shimada et al. [51]. The antioxidant capacity is expressed as Trolox equivalent. The absorbance at 517 nm was determined by spectrophotometer UV-2401 PC. In goji leaves and fruit extracts, polyphenol identification in UPLC-PDA-ESI-MS/MS (ultra-performance liquid chromatography with photodiode array and electrospray ionization tandem mass spectrometry detection) was executed using an ACQUITY Ultra Performance LC system appointed with a binary solvent manager, a photodiode array detector (Waters Corporation, Milford, MA, USA) and a G2 Q-TOF micro mass spectrometer (Waters, Manchester, UK) equipped with an electrospray ionization (ESI) source operating in both negative and positive modes [52].

### 3.5. Antidiabetic Activity (α-Amylase, α-Glucosidase)

The activity of the parasite and host extracts was assayed according to the procedure described previously by Podsedek et al. [35] (α-glucosidase) and Nickavar and Yousefian [53] (*α*-amylase). All samples were assayed in triplicate and the result was expressed as IC50. The amount of the inhibitor (expressed as mg per 1 mL of the reaction mixture under assay conditions) required to inhibit 50% of the enzyme activity is defined as the IC50 value. All samples were assayed in triplicate.

### 3.6. Antimicrobial Activity Assay

The antimicrobial activity of the extracts was evaluated using the agar well diffusion method [54]. Ready-made sterile Petri dishes (Ø 90 mm) with a Mueller–Hinton medium were used. The cell suspensions (100 µL) were evenly distributed on the Petri dishes. Six-millimeter wells were punched into the agar with a sterile Pasteur pipette, in which 60 µL of the extracts was applied. Gentamycin was used as a control for bacteria. The dishes were incubated at 37 °C for 24 h. The antimicrobial activity was evaluated by measuring the diameter of the circular inhibition zones around the well.

For the evaluation of the antimicrobial activity, the following were used: Gram-positive bacteria *Staphylococcus aureus* (ATCC-25923), *Bacillus subtilis* (ATCC-12228), and *Listeria monocytogenes* (ATCC-19115), and Gram-negative: *Escherichia coli* (ATCC-25922) and *Proteus vulgaris* 458. The MIC (Minimal Inhibitory Concentration) of the solutions was determined for each strain. The MIC value determines the lowest oil concentration (i.e., 10, 25, 50, 75, and 100 μg/mL), at which no growth (turbidity) of the tested bacterial strain is visually observed, and at the same time precedes the concentration at which growth is visible. For the above determinations, methodology consistent with Inouye et al. [55] was used.

### 3.7. Area of one Leaf (cm^2^)

In autumn, the foliage area (taken from the central part of annual shoots) was measured using the Delta Image Analysis System (Delta-T Devices LTD, England) scanner connected to the computer.

### 3.8. Color and Pigment Parameters

The color parameters assessed were L* (L* = 100 means white; L* = 0 means black), *a** (+*a** means redness; −*a** means greenness), *b** (+*b** means yellow; −*b** means blue). Measurements were obtained with an aperture diameter of 3 mm; color was measured in glass cuvettes, through a 10° observer type and D65 illuminant. CIE L**a*b** (Color Measurement Committee of the Society of Dyers and Colorists) was measured using a spectrophotometer (Konica Minolta CM-700d) [56,57]. The pigment contents are displayed on the screen as normalized difference vegetation index (NDVI) and normalized anthocyanin index (NAI) [58]. The dried plant material was powdered in a laboratory mill in triplicate. About 3 g of ground plants were poured into the glass cuvette, and 50 measurements were made in triplicate. The samples were mixed before each measurement.

### 3.9. Statistical Analysis

All statistical analyses were performed using Statistica 13.0 (StatSoft Polska, Cracow, Poland). Non-parametric methods (Kruskal–Wallis test) were used if neither the homogeneity of variance nor the normality of distribution had been previously established. The statistical significance of the differences between means was determined by testing the homogeneity of variance and normality of distribution, followed by ANOVA with Tukey’s post hoc test. The significance was set at *p* < 0.05. To determine the relationship between the cultivars and macro- and microelements, the results obtained were subjected to agglomerative cluster analysis and classified into groups in a hierarchical order by means of the Ward’s method. Multivariate analysis was performed by applying principal component analysis (PCA). The data were auto-scaled during pre-processing.

## 4. Conclusions

The obtained results enrich the knowledge of the composition and nutritional values of fresh goji fruit grown in northeast Europe and will help to verify the information given on the packaging. Differences in the composition of the macro- and microelements of the two goji cultivars No. 1 and New Big, grown in an orchard and under *in vitro* conditions were shown. These differences resulted mainly from the growing conditions of these shrubs and the composition of the soil. Goji berries cultivated *in vitro* were confirmed as an important source of healthy compounds, providing a significant contribution to the diet through both its fruit and leaves. The cultivars were rich in macro- and microelements and low in levels of toxigenic elements (i.e., Pb and Ni)

The color of the leaves and the NDVI indicated that the plants had optimal nutrient content, so it can be concluded that the assigned norms of the macro- and microelements can be a good indicator of their nutrition.

The results presented in this study provide information that goji berries grown in Poland are an interesting fruit in terms of their important health-promoting contents such as macro- and microelements, antioxidants, and anti-microbiological properties, sugars, organic acid, and phenolic acids.

Among the Gram-positive bacteria, *B. subtilis* proved to be the most sensitive to the extracts, and among the Gram-negative bacteria, it was *P. vulgaris*. The strongest inhibitory and bactericidal effect (i.e., the lowest MIC values) in relation to the majority of the examined bacteria was found in the extract from the ‘New Big’ leaves collected in the orchard. Therefore, it can be concluded that the New Big cultivar is a source of active substances that inhibit the growth and development of selected types of bacteria.

The knowledge obtained from this study will help determine the commercial potential of goji berries used for nutraceutical applications and of the incorporation in food preparation that improves human health. Taking into consideration the RDAs for minerals established by the EU, fresh goji fruit can be a source of Cu, Fe, Mn, Zn, P, and Se.

## Figures and Tables

**Figure 1 molecules-25-05314-f001:**
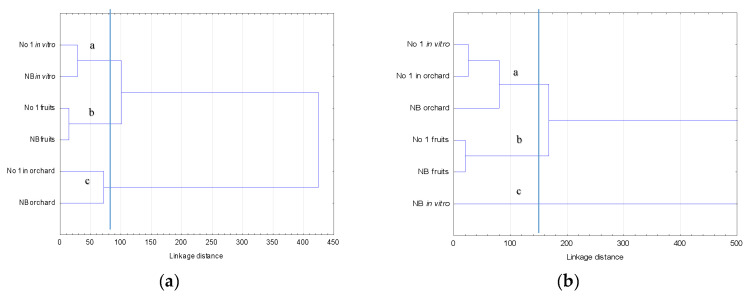
Dendrogram of cluster analysis of micro- and microelements (**a**) and polyphenols (**b**) in two cultivars of goji fruit and leaves (cut off—85 and 150).

**Figure 2 molecules-25-05314-f002:**
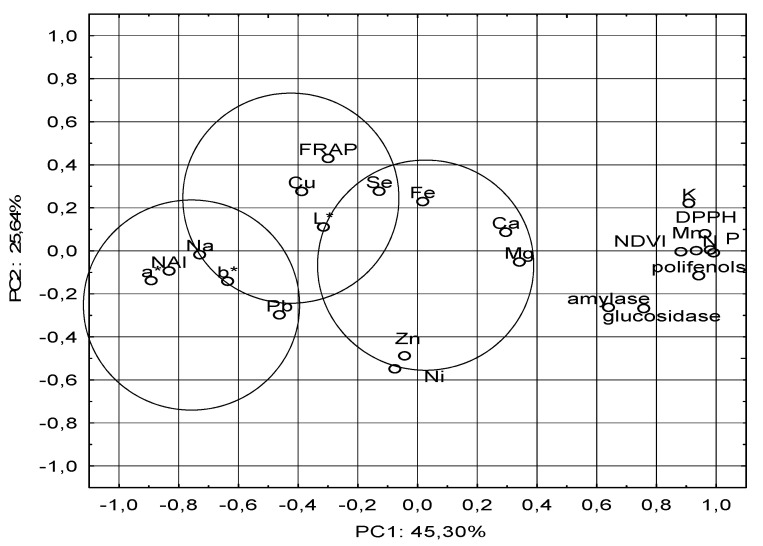
The principal component analysis (PCA) for mineral compositions and phytochemical content of goji leaves and fruit depending on growth conditions.

**Table 1 molecules-25-05314-t001:** Average values of macroelements in leaves and fruit of two cultivars of goji cultivated in orchard and *in vitro* condition.

Compounds (g/kg)	Leaves				Fruit	
	No. 1		New Big		No. 1	New Big
	Orchard	*In Vitro*	Orchard	*In Vitro*	Orchard	
N (22–32 ^2^)	42.08 ± 1.71a ^1^	57.14 ± 2.37b	43.50 ± 1.98a	77.92 ± 3.05c	29.85 ± 1.30B	24.32 ± 0.92A
P (19–30)	7.85 ± 0.42a	11.93 ± 0.56c	8.52 ± 0.47b	13.78 ± 0.51b	5.38 ± 0.19B	4.85 ± 0.15A
K (12–20)	29.57 ± 1.17b	57.06 ± 1.98d	14.73 ± 0.52a	52.06 ± 1.50c	4.30 ± 0.13A	3.98 ± 0.11 A
Ca (4–8)	11.00 ± 0.52b	3.87 ± 0.18a	11.39 ± 0.48b	4.00 ± 0.21a	0.91 ± 0.04B	0.75 ± 0.03A
Mg (2–4.4)	5.89 ± 0.17b	2.87 ± 0.09a	7.50 ± 0.27c	3.04 ± 0.11a	1.11 ± 0.05A	1.02 ± 0.05A
Na (no data)	4.22 ± 0.23c	2.17 ± 0.19b	4.60 ± 0.25d	1.82 ± 0.13a	4.03 ± 0.19B	3.82 ± 0.15A

^1^ Means followed by the same letter in lines do not differ significantly at P = 0.05 according to Tukey multiple range/small letters for leaves, capital–fruit. ^2^ Optimal content for leaves according to Glonek and Komosa [15].

**Table 2 molecules-25-05314-t002:** Average values of microelements in leaves and fruit of two cultivars of goji cultivated in orchard and *in vitro* conditions.

Compounds (g/kg)	Leaves				Fruit	
	No. 1		New Big		No. 1	New Big
	orchard	*In vitro*	Orchard	*In vitro*	Orchard	
Fe (40–60 ^2^)	120.13 ± 7.20c ^1^	71.00 ± 3.82a	97.81 ± 4.07b	77.44 ± 3.55a	66.03 ± 2.04A	79.44 ± 2.63B
Zn (8–14)	18.62 ± 0.35b	26.77 ± 0.47c	14.53 ± 0.27a	35.07 ± 0.31d	8.16 ± 0.24A	8.73 ± 0.19B
Mn (70–260)	50.70 ± 1.04b	176.89 ± 3.57c	42.07 ± 0.88a	244.64 ± 5.03d	7.04 ± 0.12A	7.74 ± 0.10B
Cu (5–20)	8.42 ± 0.19d	3.39 ± 0.16a	6.06 ± 0.14c	3.90 ± 0.11b	4.72 ± 0.98A	5.64 ± 1.25B
Se	0.089 ± 0.005b	n.d. ^3^	0.062 ± 0.004a	n.d.	0.012 ± 0.001B	0.009 ± 0.001A
Pb	0.034 ± 0.03a	n.d.	0.054 ± 0.05b	n.d.	0.017 ± 0.001A	0.029 ± 0.002B
Cd	n.d.	n.d.	n.d.	n.d.	n.d.	n.d.
Ni	0.019 ± 0.002b	n.d.	0.011 ± 0.002a	n.d.	0.007 ± 0.001B	0.003 ± 0.000A

^1^ Designation according to Table 1. ^2^ Optimal content for leaves according to Glonek and Komosa [15]. ^3^ n.d.—not detected.

**Table 3 molecules-25-05314-t003:** The leaf area, color measurement and NAI and NDVI value in leaves and fruit of two cultivars of goji cultivated in orchard and *in vitro* condition.

Compounds		Cultivar					
		No. 1		New Big		No. 1	New Big
		Leaves				Fruit	
		Orchard	*In Vitro*	Orchard	*In Vitro*	Orchard	
Leaf area (cm^2^)		3.53 ± 0.31c ^1^	2.11 ± 0.17a	4.18 ± 0.39d	2.78 ± 0.15b	-	-
Color parameters	L*	47.82 ± 2.55a	42.36 ± 1.53b	42.50 ± 2.71b	34.57 ± 1.70c	35.88 ± 1.45B	44.72 ± 2.04A
	*a**	−36.27 ± 3.08ab	−38.94 ± 1.98b	−33.84 ± 2.55a	−43.56 ± 2.12c	25.63 ± 3.24A	33.34 ± 3.42B
	*b**	30.63 ± 2.50c	27.73 ± 1.45bc	23.71 ± 2.26a	25.25 ± 1.63ab	27.05 ± 2.44A	41.40 ± 2.79B
NAI		−0.76 ± 0.10b	−0.69 ± 0.06a	-0.82 ± 0.07c	−0.74 ± 0.05b	0.62 ± 0.05A	0.71 ± 0.05B
NDVI		0.69 ± 0.04a	0.76 ± 0.05b	0.84 ± 0.05c	0.91 ± 0.03d	−0.46 ± 0.04B	−0.38 ± 0.03A

^1^ Designation according to Table 1.

**Table 4 molecules-25-05314-t004:** Content of polyphenolic compounds in leaves and fruit of two cultivars of *L. chinensis*.

Compounds (mg/100 g DW)	Leaves				Fruits	
	No. 1		New Big		No. 1	New Big
	*In Vitro*	Orchard	*In Vitro*	Orchard		
Quercetin-3-*O*-Gal	n.d.	n.d.	n.d.	n.d.	0.80A ^1^	2.98B
Kaempferol-3-*O*-Glc-7-*O*-Soph	n.d.	n.d.	11.47a	14.25b	n.d.	n.d.
Quercetin-3-*O*-Rut-7-*O*-Glu	n.d.	n.d.	n.d.	n.d.	2.94B	2.07A
Quercetin-3-*O*-Soph-7-*O*-Rha	12.73b	14.65b	4.42a	24.03c	0.24A	2.31B
Kaempferol-3-*O*-Rut-7-*O*-Glu	n.d.	n.d.	n.d.	n.d.	6.38A	17.22B
Quercetin-3-*O*-Glu	n.d.	n.d.	2.37b	0.72a	3.05A	3.50A
Quercetin-3-*O*-Rut	12.58a	23.81b	31.42c	33.89c	11.29B	7.57A
Quercetin-3-*O*-Glu-7-*O*-Rha	0.22a	0.48b	1.04c	0.58b	n.d.	n.d.
Kaempferol-3-*O*-Rhu	0.77a	0.95b	n.d.	n.d.	0.18	n.d.
Kaempferol-3-*O*-Glu-7-*O*-Rha	0.68a	0.75a	1.21c	0.94b	n.d.	n.d.
*Total flavonols*	*26.98a*	*40.64b*	*51.93c*	*74.41d*	*24.88A*	*35.65B*
Procyanidin B dimer	0.99b	3.04c	0.49a	8.01d	0.62A	0.51A
(+)-Catechin	n.d.	n.d.	7.38a	21.41b	18.44B	11.04A
*Total flavan-3-ols*	*0.99a*	*3.04b*	*7.87c*	*29.42d*	*19.06B*	*11.55A*
Tetragalloyl-glucose	0.23a	0.29a	4.22c	3.81b	n.d.	n.d.
Galloylquinic acid	0.17a	0.21a	2.55c	2.09b	n.d.	n.d.
Total hydrolyzable tannins	0.40a	0.50a	6.77c	5.9b		
5-*O*-Ferruloylquinic acid	0.41b	0.30a	1.77	1.60c	0.56B	0.21A
p-Coumaric acid	1.66	1.52	2.15d	1.06	8.29B	6.35A
Caffeic acid	0.64b	0.82c	0.37a	0.34a	n.d.	n.d.
Caftaric acid	n.d.	n.d.	n.d.	n.d.	0.74A	5.06B
p-Coumaroyl acid dihexoside	n.d.	n.d.	n.d.	n.d.	4.22B	3.50A
3-*O*-Caffeoylquinic acid (neochlorogenic acid)	0.51c	0.56c	0.38b	0.27a	4.11A	11.04B
3-*O*-Caffeoylquinic acid derivative	1.33d	1.12c	0.17a	0.33b	10.24B	8.22A
4-*O*-Caffeoylquinic acid (cryptochlorogenic acid)	0.44b	0.58c	0.20a	1.87d	n.d.	n.d.
5-*O*-Caffeoylquinic acid (chlorogenic acid)	66.47c	73.05d	31.04a	52.15b	n.d.	n.d.
5-*O*-Caffeoylquinic acid isomer	2.81d	2.33c	1.27b	0.44a	2.52A	4.21B
*Total phenolic acids*	*71.56c*	*77.64c*	*33.06a*	*55.06b*	*21.09A*	*26.97A*
TOTAL	102.64A	124.46B	103.92A	167.79C	74.62A	85.79B

^1^ Designation according to Table 1.

**Table 5 molecules-25-05314-t005:** Health-promoting capacities of leaves end fruit of two cultivars *L. chinensis*.

Nutritional Value		Leaves				Fruit	
		No. 1		New Big		No. 1	New Big
		*In Vitro*	Orchard	*In Vitro*	Orchard	Orchard	
DPPH (mmol Trolox/100g)		3.88c^1^	3.26b	4.25d	2.48a	7.61B	5.33A
FRAP (mmol Trolox/100g)		4.02c	2.54a	5.89d	3.48b	2.89A	3.93B
*α*-amylase IC50 (mg/mL)		112.6b	75.1a	172.9d	134.0c	33.45A	37.01A
*α*-glucosidase IC50 (mg/mL)		25.41a	22.05a	42.28b	37.06b	8.36A	7.44A
Soluble sugars (g/100 g DW)	fructose	1.88c	0.89a	2.51d	1.33b	9.67A	11.83B
	glucose	1.55c	0.51a	1.88d	1.04b	10.11A	13.06B
	sucrose	0.20b	0.24c	0.18b	0.12a	0.51A	0.77B
Organic acid (g/100 g DW)	oxalic acid	0.021a	0.024a	0.073b	0.080b	n.d.	0.362
	citric acid	0.115b	0.292c	0.083a	0.314d	0.951A	1.485B
	succinic acid	0.027ab	0.035bc	0.021a	0.042c	0.547B	0.322A
	fumaric acid	0.019bc	0.023c	0.016ab	0.014a	0.078B	0.066A

^1^ Designation according to Table 1.

**Table 6 molecules-25-05314-t006:** Antibacterial activity and Minimal Inhibitory Concentration of leaves extracts and antibiotics against bacterial species tested by disc diffusion assay.

	Inhibition Zone—IZ (mm) and Minimal Inhibitory Concentration—MIC (μg/mL)								
Bacterial Strains	No. 1				New Big				Standard Antibiotic (Gentamicin)
	Orchard		*In Vitro*		Orchard		*In Vitro*		
	IZ	MIC	IZ	MIC	IZ	MIC	IZ	MIC	
*E. coli*	9.6 ± 0.9	>100	8.6 ± 0.6	>100	10.5 ± 0.8	100	9.8 ± 0.8	>100	12.5
*P. vulgaris*	12.2 ± 0.8	75	11.3 ± 0.9	75	14.3 ± 0.8	50	13.5 ± 0.9	100	17.0
*B. subtilis*	14.7 ± 0.8	100	13.2 ± 1.0	100	16.7 ± 1.1	100	14.2 ± 0.8	>100	21.4
*S. aureus*	9.0 ± 0.6	75	8.8 ± 0.9	100	9.6 ± 0.8	75	9.1 ± 0.7	>100	14.6
*L. monocytogenes*	10.3 ± 0.8	100	10.2 ± 0.7	>100	10.9 ± 0.9	75	10.5 ± 0.4	>100	18.6

**Table 7 molecules-25-05314-t007:** Mineral composition of soil in which the two goji cultivars grew in orchard.

N	P	K	Mg	Ca	Na	Fe	Mn	Zn	Cu	Cd	Pb	Ni	Se
g/kg	mg/kg												
17.33	123.3	284.7	72.7	452	11.3	83.4	57.3	37.0	7.52	0.296	31.3	4.67	0.031

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
