# Peer review of "Health-Promoting Capacities of In Vitro and Cultivated Goji (Lycium chinense Mill.) Fruit and Leaves; Polyphenols, Antimicrobial Activity, Macro- and Microelements and Heavy Metals"

_molecules, 2020, doi:10.3390/molecules25225314_

Round 1
Reviewer 1 Report
The theme of the article is very interesting. It deals with health promoting capacities of plant materials which are investigated intensively in the last few decades and therefore this paper gives an interesting and useful data.
Manuscript is readable in appropriate length, relatively good structured and with enough cited references.
But, there are some minor errors in a manuscript, what should be corrected according to Instructions for authors. For example:
- In the text, reference numbers should be placed in square brackets [ ].
- In the references list references must be numbered in order of appearance in the text.
- The references in the references list should be described according to journal instructions for authors (Please read “MDPI Reference List and Citations Style Guide”)
- There are some minor mistakes in spelling that should be corrected (please use superscript numbers when you talk about surface area (line 215)…..
Besides that, research methodology is correct and quite well applied. In chapter ‘Results and discussion’ obtained results are clearly explained and correctly compared with other authors investigations. The conclusions are justified by the data.
This manuscript will be acceptable for publication after minor revision.
Reviewer 2 Report
Dear Authors
In this study, authors two cultivars of goji were propagated in vitro and grown in an orchard and were evaluated for macro- and microelements and harmful heavy metals. The nutritional value, polyphenols and the antimicrobial activity were also assessed. Results showed that the leaves have higher content of polyphenols comparing to the fruit. The fruit were characterized by health promoting capacities, while the leaves had antibacterial activity. Finally, they conclude that goji berries can be declared as a source of Cu, Fe, Mn, Zn and P according to the Recommended Daily Allowance (RDA) for minerals. This report is comprehensively described and the conclusions appear sound. However, some data is not well organized, the results did not fit the conclusion, and some discussion need to be address. Moreover, some figures and tables are need to be redraw, and English language and style need to be changed. This manuscript can be reconsidered after major revision.
In figure 1, authors conducted a cluster analysis by using Ward’s method. They concluded that results permitted the isolation of two groups with a similar influence on the micro- and macro elements in goji fruit. However, according to the Figure 1a and 1b, the results did not show the similar influence on both different elements. Please discuss the differences between them. In addition, please assign the horizontal axis title in English.
In Fig. 2, authors analyzed the mineral compositions and phytochemical content of goji leaves and fruit depending on growth conditions by PCA method. They found a high correlation between polyphenols and NDVI, between FRAP and some microelements, especially Fe, between NAI index and color parameters a* and b*. However, authors did not discuss those results in detail with particular references. What is the relationship of NDVI and polyphenol? FRAP and Fe? In addition, the figure is too small to read and certain group should be draw in circle to separate the different groups.
Line 440, in materials and methods, table 1 should be label as table 8.
In references citation, the authors did not follow the rule of the “Molecules” journal. References must be numbered in order of appearance in the text (including citations in tables and legends) and listed individually at the end of the manuscript. In the text, reference numbers should be placed in square brackets [ ], and placed before the punctuation; for example [1], [1–3] or [1,3]. For embedded citations in the text with pagination, use both parentheses and brackets to indicate the reference number and page numbers; for example [5] (p. 10), or [6] (pp. 101–105).
Reviewer 3 Report
This is an interesting manuscript dealing with a complete characterization of In Vitro and 2 cultivated Goji (Lycium chinense Mill.) fruits and leaves. Multiple parameters were addressed and, in general, the results were correctly discussed and compared with those described in the literature for the same samples or similar ones. In my opinion, this manuscript is of interest to the scientific community dealing with the important field of food characterization, and the manuscript can be accepted for publication after minor revision.
- The results and discussion section contains information that is not based on results but could be included in the introduction section. For example:
Lines 82-93. This explanation could perfectly be placed in the introduction. This is information not based on the results. This is only one example, similar situations are found through the manuscript.
- Lines 197-200: “the results of this study are interesting and very helpful and important”. This is a general attribution to any study of the same characteristics; these 3-4 lines can be rewritten and also placed in the introduction section.
- The authors need to revise the citation requirements for Molecules. I believe that references must be introduced within the text as [1], and not by the author names and years.
- Line 272: Please indicate correctly the method employed. Defined as UPLC-PDA-ESI-MS/MS there is no idea of the MS instrument used. Your method is based on Ref. 25, so please indicate also that a TOF instrument was employed, because this will give more confidence in the identification capabilities of the proposed methodology.
- “SUME” in Table 5?
- Figure 2. Please indicate that you are showing the PCA loading plots. It may be interesting to see the combined plot (scores and loadings).
- Finally, there are some English mistakes through the manuscript so I recommend its revision.
Round 2
Reviewer 2 Report
The manuscript has been improved in detail as we suggested. Authors have responded to all comments and made changes in the text. We recommend this manuscript to be accepted for publication.